# The Consensus Problem in Polities of Agents with Dissimilar Cognitive Architectures

**DOI:** 10.3390/e24101378

**Published:** 2022-09-27

**Authors:** Damian Radosław Sowinski, Jonathan Carroll-Nellenback, Jeremy DeSilva, Adam Frank, Gourab Ghoshal, Marcelo Gleiser

**Affiliations:** 1Thayer School of Engineering, Dartmouth College, Hanover, NH 03755, USA; 2Department of Physics and Astronomy, University of Rochester, Rochester, NY 14627, USA; 3Department of Anthropology, Dartmouth College, Hanover, NH 03755, USA; 4Department of Physics and Astronomy, Dartmouth College, Hanover, NH 03755, USA

**Keywords:** agent, polity, transfer entropy, information theory, consensus, sociophysics

## Abstract

Agents interacting with their environments, machine or otherwise, arrive at decisions based on their incomplete access to data and their particular cognitive architecture, including data sampling frequency and memory storage limitations. In particular, the same data streams, sampled and stored differently, may cause agents to arrive at different conclusions and to take different actions. This phenomenon has a drastic impact on polities—populations of agents predicated on the sharing of information. We show that, even under ideal conditions, polities consisting of epistemic agents with heterogeneous cognitive architectures might not achieve consensus concerning what conclusions to draw from datastreams. Transfer entropy applied to a toy model of a polity is analyzed to showcase this effect when the dynamics of the environment is known. As an illustration where the dynamics is not known, we examine empirical data streams relevant to climate and show the *consensus problem* manifest.

## 1. Introduction

Agent success necessitates predicting with fidelity the behavior of treacherous environments in order to optimize action. Information theory is the power-tool for quantifying predictability, so any agent that thinks—an *epistemic* agent—can be modeled as a computing entity attempting to infer information theoretic measures. Channel capacity, Shannon entropy, mutual information, transfer entropy, and other such measures are defined with respect to empirically inaccessible joint probability distributions [1,2,3,4]. Finite computational and sensory resources, such as memory and sampling frequency, respectively, affect the estimation of these distributions [5,6,7,8,9,10,11,12,13,14,15,16,17]. From a Bayesian perspective, this implies that any agent with finite cognitive resources will make judgements that are governed by such limitations [18,19,20,21]. A group of epistemic agents—a *polity*—predicated on the pooling and dissemination of information is affected by the architectural heterogeneity of its constituents. In this context, a question of interest is to what extent can agreement be reached within polities given such limitations?

Transfer entropy provides a canonical example of how architecture affects conclusions about the structure of the environment. It is well known that sub-sampling of continuous time series can lead to differing estimates of transfer entropy [22,23,24], a failure circumvented by considering rates in the continuum limit Δt→0 [10,24]. Mathematically satisfying, such considerations are not applicable to physical agents whose information storage and processing capabilities are limited by the universe they find themselves attempting to predict and navigate [25]. Szilard and Landauer revealed that information is physical; there is an energy cost in memory for the read-write cycle [26,27,28]. Bekenstein demonstrated that any finite volume of spacetime has an upper limit to the information it can contain [29,30,31]. Bremermann showed that information cannot be processed at any rate without loss of fidelity [32]. Herein these limitations are not swept under the rug—any finite agent must have computational/cognitive abilities limited by finite memory and finite sampling frequency, which we bundle together under the term *cognitive architecture*. Embracing the inevitability of differing estimates of information theoretic measures due to variance in cognitive architectures or allotment of cognitive resources gives a different perspective on the origins of disagreement—the Consensus Problem.

With the desire to shed some light on what this means, we introduce a toy model of a polity, built from the bottom up, in order to show how disagreement amongst agents can emerge in ideal circumstances. We start with an epistemic agent capable of harnessing the correlations in the history of its environment to predict the future. The agent is placed in a simple environment consisting of a pair of stimuli which it samples according to its own limitations. We develop an exactly solvable model for the transfer entropy between the two stimuli and analyze how the inference of information-flow depends on the memory-capacity and temporal sampling of an observing agent [22,23,24]. Finally, a polity of agents, each having access to identical streams of stimuli, is formed and their individual estimates brought together and compared. As a representative illustration, we use time-series of CO${}_{2}$ content and atmospheric temperature taken on Mauna Loa to demonstrate this effect, suggesting that polarization amongst epistemic agents on scientifically charged topics may persist irrespective of the quality of data supporting one particular conclusion.

## 2. From Agents to Polities

In this section, we construct a model of an epistemic agent. We emphasize *epistemic* since we are not concerned with the actions of the agent, but in how it uses memories of its environment to predict the future and inform itself of how it *should* act. The agent is placed into an environment in which it has access to two time series of sensory input, which it samples as coupled qualia and stores in memory. Both sampling rate and storage are fixed—a toy model of the intrinsic cognitive architecture of the agent. We give a brief primer on transfer entropy and then construct a toy model of the qualia coming from a complex environment. With a single EA in hand, we introduce a simple description of a *polity*, an ensemble of epistemic agents, and show how the consensus problem can emerge in heterogeneous populations of agents.

### 2.1. Transfer Entropy and Influence

An epistemic agent, simple or complex, exists in a world full of uncertainty. That uncertainty implies that the inference of probability distributions, and the measures derived from them, lies at the core of an agent’s model of the world. Physical limits on processing and/or reaction times can hamper the actions, and existence, of an agent if that model only describes the *past* states of its environment. A model of the *future* helps mitigate the vagaries of the world.

Transfer entropy [4,33] is the simplest measure quantifying the extent to which past correlations between two processes reduce the future uncertainty in either process—the extent to which an agent can predict one of the processes, given past knowledge of both. Given two processes, $X_{1}\left(t\right)\wedge X_{2}\left(t\right)$, the transfer entropy is defined as the excess information gained by knowing the past history of process 2 in addition to that of process 1,
(1)$$\begin{array}{c} T_{2\to 1}\;=\;H\left[X_{1},t|X_{1}^{\prime },t^{\prime }\;<\;t\right]\;-\;H\left[X_{1},t|X_{1}^{\prime }\wedge X_{2}^{\prime },t^{\prime }\;<\;t\right], \end{array}$$
where H[X] is the hidden information, i.e., the Shannon entropy of *X*, and H[X|Y] is the conditional entropy of *X* conditioned on *Y* [1]. In what follows, it is useful to rewrite the transfer entropy in terms of the mutual information M[X:Y], expressed as the Kullback–Leibler divergence between joint and product distributions, $M\left[X:Y\right]=D_{KL}(\rho_{XY}\left|\right|\rho_{X}\;⊗\;\rho_{Y})$ [34]. Using this, we can write an alternative expression for the transfer entropy [2,3,35]:(2)$$\begin{array}{c} T_{2\to 1}\;=\;M\left[X_{1},t\;:\;X_{1}^{\prime }\;\wedge \;X_{2}^{\prime },t^{\prime }\;\;<\;t\right]\;-\;M\left[X_{1},t\;:\;X_{1}^{\prime },t^{\prime }\;\;<\;t\right]. \end{array}$$
For any finite agent, memory limitations preclude storage of the entire history, so a discrete subset of events is taken at $t^{\prime }\in {\left\{t-n\Delta t\right\}}_{n=1,\ldots ,N}$, where *N* is taken to represent the number of past snapshots of the world the agent stores in estimating the joint distribution over histories i.e., memory. A schematic of the information diagram is shown in Figure 1, highlighting the relevant measures.

The processes in question can be treated symmetrically—how does knowing how $X_{2}$’s history decreases the uncertainty in $X_{1}$’s future compare to knowing how $X_{1}$’s history reduces the uncertainty in $X_{2}$’s future? To probe the asymmetry of transfer entropy in these two cases, it is useful to define the ratio of the difference in information flows to the total information flow:(3)$$\begin{array}{c} T_{ij}=\frac{T_{i\to j}-T_{j\to i}}{T_{i\to j}+T_{j\to i}}. \end{array}$$
This quantity is bounded on the interval [−1,1], saturating iff the transfer entropy vanishes in only one direction, which occurs iff the target process is deterministic. Since $T_{i\to j}\geq 0$, when both directions vanish we set $T_{ij}\equiv 0$. We note that, in certain cases, the transfer entropy reduces to the Granger-*Causality* [36,37], and there is a tendency to interpret it as a causal measure. However, one must be cautious employing such interpretations, given that both measures are based entirely on correlations [36,38,39,40]. In passing, we refer to it as *influence*, but only with regard to how correlations help influence an agent’s predictions, as opposed to any sort of causal influence between the actual processes.

### 2.2. A Toy Model of Qualia

Consider now a simple agent, one whose senses allow it to *observe* the world through two stimuli. The experiences of these stimuli by the agent are referred to as qualia (singular quale); both terms are used interchangeably. Placing the agent into an environment provides it with these stimuli, albeit noised by whatever else is happening. One could have in mind a thermostat that senses both temperature and pressure, though, in principle, one can imagine a far more complex agent sampling myriad stimuli and ignoring all but two. Like the temperature and the pressure of the thermostat’s environment, the stimuli are coupled together by whatever processes give rise to them in the environment, and the agent will estimate the influence of the two by computing the transfer entropy from its memory and sampling of the environment.

To see this in action, let us consider as our stimuli a pair of positions, $X_{1}\left(t\right)\wedge X_{2}\left(t\right)$, coupled in the environment by evolution equations
(4)$$\begin{array}{cc} \frac{dX_{1}}{dt} & =-\alpha_{1}\left(X_{1}-X_{2}\right)+\beta_{1}\eta_{1}, \end{array}$$
(5)$$\begin{array}{cc} \frac{dX_{2}}{dt} & =-\alpha_{2}\left(X_{2}-X_{1}\right)+\beta_{2}\eta_{2}, \end{array}$$
and initial conditions $X_{1}\left(t_{0}\right)\wedge X_{2}\left(t_{0}\right)$. These could be the temperature and pressure of the thermostat, but dimensionalized to facilitate comparisons. Since a metric structure is closely tied to similarity judgements when comparing both auditory and visual qualia, *position* is an appropriate descriptor [41,42] Here, $\alpha_{i},\beta_{i}\in R^{+}$ parameterize the deterministic and stochastic contributions, respectively, to the equations of motion. The latter are associated with the environment acting as a heat reservoir, R, inducing fluctuations in both processes—$\eta_{1,2}$ are independent white noise contributions that satisfy $\langle \eta_{i}\left(t\right)\rangle =0$ and $\langle \eta_{i}\left(t\right)\eta_{j}\left(t^{\prime }\right)\rangle =\delta_{ij}\delta \left(t-t^{\prime }\right)$. The dimensions of the parameters are $\left[\eta_{i}\right]={\mathrm{Time}}^{-1/2}$, $\left[\alpha_{i}\right]={\mathrm{Time}}^{-1}$ and $\left[\beta_{i}\right]=\mathrm{Length}\;\times \;{\mathrm{Time}}^{-1/2}$.

The four coupling constants define natural time and length scales in the model,
(6)$$\begin{array}{c} \tau =\frac{1}{\alpha_{1}+\alpha_{2}}\;\ell =\frac{\beta_{1}+\beta_{2}}{\sqrt{\alpha_{1}+\alpha_{2}}}, \end{array}$$
which are used to dimensionalize all variables; see Appendix A for more details. The latter describes the size of fluctuations in the separation of the two processes, while the former the decay of transients. In the left panel of Figure 1, we plot instances of paths generated by Equations (Equation 4) and (Equation 5) for several values of the coupling constants. We note that the model has an explicit coupling which introduces correlations between the past and present of both processes. The relevant quantities determining the behavior of the information dynamics of the two processes are
(7)$$\begin{array}{c} a=\frac{\alpha_{1}-\alpha_{2}}{\alpha_{1}+\alpha_{2}}\;b=\frac{\beta_{1}-\beta_{2}}{\beta_{1}+\beta_{2}}, \end{array}$$
with the former being deterministic and the latter stochastic asymmetry parameters, both normalized to the range [−1,1].

### 2.3. Influence between Qualia

Equations (Equation 4) and (Equation 5) are diagonalized into a Wiener process for the center of mass motion, and an Ornstein–Uhlenbeck process for the separation, giving exact solutions. For a more in depth derivation, please see Appendix A. Given that both are Gaussian processes, the solution to the corresponding Fokker–Planck equation is a multivariate Gaussian, allowing for an exact solution for the analytical form of $T_{12}\left(\Delta t,N\right)$. For a Gaussian process, if A and B have covariances $\Sigma_{AA}$ and $\Sigma_{BB}$, respectively, and joint covariance Σ, then the mutual information is
$$M\left[A:B\right]=\frac{1}{2}\log\frac{|\Sigma_{AA}\left|\right|\Sigma_{BB}|}{\left|\Sigma \right|}.$$
Combining this with Equation (Equation 2), we show the results for different values of *N* in Figure 3. In the small, ϵ≪1, asymmetry regime where $\left(a+b\right)∼O\left(\epsilon \right),|b-a|∼O\left(\epsilon^{2}\right)$ and for small timescales, $\Delta t∼O(\epsilon^{2})$, short memory N=1, and in the limit t→∞, the influence reads
$$\begin{array}{c} T_{12}\left(\Delta t,1\right)=\frac{3}{\ln2}\left(a+b\right)\left(\Delta t-{\left(a+b\right)}^{2}\right)+\;O\left(\epsilon^{4}\right). \end{array}$$
We note that there are many ways in which one can discuss a *small* limit, and this is simply one of them. When the asymmetry switches sign, agents sampling timescales on opposite sides of $\Delta t∼{\left(a+b\right)}^{2}$ conclude in contradiction to one another the direction of influence. Since the transfer entropy for Gaussian processes is equivalent to the Granger causality, this implies that one can confuse the direction of causation, which, as previously mentioned, is a common misinterpretation of a purely correlative measure [36,37]. Figure 2 indicates that this phenomenon is not limited to the linear regime but is a general feature found across parameter space. A longer discussion of how to compute the influence within the qualia model is presented in Appendix B.

It is beneficial to think of a space of all possible cognitive architectures given fixed values of deterministic and stochastic model parameters. A polity distribution function, discussed in the next section, has this space as its support. If the population has a fairly uniform set of architectures, then this density will be highly peaked. As variance in architecture is introduced, or as cognitive resources are moved around, the polity distribution function spreads out and deforms. When the distribution crosses the vanishing influence surface, agents on either side will necessarily have differing opinions.

### 2.4. A Toy Model of Polities

A polity can be thought of as a more complex agent, one whose beliefs are a conglomeration of the beliefs of its constituent agents, and whose actions are determined by that distribution. Pooling of information can be done at multiple levels, and in different fashions that distribute weight amongst agent opinions. Here, we consider a simple case where final estimates of an information measure are pooled together but examine the effect of different weight schemes. Bringing together the estimates of many individual agents, the naive expectation is that the law of large numbers might allow the polity to form an estimate with smaller uncertainty. Given a homogeneous population of agents, this would be the case; not so for heterogeneous polities.

Consider then a polity of agents with heterogeneous cognitive architectures, $A=\{\left(N_{1},\Delta t_{1}\right),\left(N_{2},\Delta t_{2}\right),\ldots \}$, described by a population distribution function, $\rho_{N}\left(\Delta t\right)d\Delta t$, giving the fraction of the population with memory *N* and sampling times between Δt and Δt+dΔt. Each agent, assumed independent of the others, has access to the same data, sampling and remembering as their nature allows, and estimates the influence between processes, contributing that result to the polity. The distribution of estimates is
$$\rho \left(T_{12}\right)=\sum_{N=1}^{\infty }\int_{0}^{\infty }\;\;d\Delta t\;\rho_{N}\left(\Delta t\right)\rho \left(T_{12}|N,\Delta t\right),$$
where an agent’s uncertainty in their estimate of $T_{12}$ can be included in the conditional distribution $\rho (T_{12}|N_{a},\Delta t_{a})$. More details on the derivation are given in Appendix B.

The distribution $\rho_{N}\left(\Delta t\right)$ describes the heterogeneity of cognitive architectures found in the polity. If the agents are similar enough that they all have near identical hardware, whether that be biological or otherwise, one would expect the distribution to be peaked at some $\Delta t_{*}$, with a large variance. To account for several orders of magnitude in sampling rates, we take the Δt dependence to be log-normal, with mean 〈Δt〉 and log normalized root mean square $\theta =\ln\sqrt{\langle \Delta t^{2}\rangle /{\langle \Delta t\rangle }^{2}}$. Meanwhile, the *N* dependence is taken to be geometric, with mean 〈N〉,
(8)$$\rho_{N}\left(\Delta t\right)=\frac{e^{-\frac{\theta }{4}}}{\sqrt{4\pi \theta }\langle N\rangle \langle \Delta t\rangle }{\left(1-\frac{1}{\langle N\rangle }\right)}^{N-1}{\sqrt{\frac{\langle \Delta t\rangle }{\Delta t}}}^{3-\frac{1}{\theta }\ln\sqrt{\frac{\langle \Delta t\rangle }{\Delta t}}},$$
where Δt∈(0,∞) and N≥1. Correlations between sampling and memory representing constraints such as favored look-back window times, 〈T〉=〈NΔt〉≠〈N〉〈Δt〉, can easily be incorporated into this framework.

## 3. Results

In this section, we look at the influence measure across a wide range of parameters in our coupled qualia model, as well as differing cognitive architectures. The central contour plot (A) of Figure 2 shows the asymmetry across the full domain for the deterministic parameter, a∈[−1,1] as inferred by agents with a range of sampling timescales, $\Delta t\in [10^{-3},10^{3}]$. In the bottom-left surface plot (A) of Figure 2, we see the corresponding transfer entropy from process 1 to process 2 (blue), and vice versa (pink). The dominating process is the one with the weaker deterministic coupling; the other process is pulled towards it, as seen in the left panel of Figure 1. Three slices, labelled I,II, and III, are taken of these surfaces for constant values of the deterministic asymmetry parameter, namely a∈{−0.1,0.2,0.5} and displayed in the upper-left three panels (C). For the first (last) of these, the transfer entropy in the direction 2→1 (1→2) is always larger. In both cases, agents would interpret the information flow as always being unidirectional, irrespective of their sampling times Δt. In the middle panel, however, there is a cross-over of transfer entropy at a particular value of temporal discretization. The exact value of Δt at which this occurs is not important in our discussion, as the far right plot of the influence shows that there is a wide range of deterministic asymmetry parameters that share this feature. In particular, the direction of influence inferred by agents will depend on their sampling of the past: If the agent samples the processes at longer timescales, they will believe that process 2 holds more information about process 1 than the other way around. Conversely, for shorter sampling timescales, the agents will reach the opposite conclusion. Re-framing our discussion to an ensemble of agents that do not have an agreed upon sampling timescale, there does not exist a singular conclusion concerning information flows across the entire ensemble: samples of agents drawn from this ensemble will reach contradictory conclusions concerning historical correlations between the processes.

The influence cross-over is not simply dependent on the sampling timescale Δt, but also the memory size of the agents captured by the parameter *N*. The plot array (D) in Figure 2 shows multiple instances of the central contour plot (A) for varying values of the noise asymmetry parameter, *b*, and memory size, *N*. The black line in these panels indicates the region where the influence flips, and the rows show its dependence as a function of increasing *N*. Since the crossover region is monotonic in *N*, there are points of constant asymmetry parameters and temporal discretization that nonetheless lead to contradictory conclusions due to different memory capacities. We note that the effect of *N* appears weaker than that of Δt as the locus of influence flips changes with *N* and appears to saturate by N∼8.

One could imagine that the way *N* and Δt affect the transfer entropies conspire together so that, for a constant look-back window, T=NΔt, their effects would cancel out. This is not the case, however, as seen in Figure 3, where the top panel (A) shows $T_{12}$ for a wide range of $T\in [10^{-2},10^{2}]$ and $N\in \{1,2,\ldots ,10^{2}\}$, and the bottom panel (B) explores the space of deterministic and stochastic parameters. The black line represents the cross-over point for the asymmetry in transfer entropy, a general feature over a large number of samples. It is interesting to note that, for a≈b, the cross-over curve hugs the line (horizontal, dashed) N=2 across window sizes smaller than the natural timescale, τ, while for larger windows it hugs the line (diagonal, dashed) corresponding to discretizations of the window into increments of size τ. As *b* grows larger than *a*, the crossover curve moves to the right, yet remains approximately parallel to this latter line for N≳2.

### The Consensus Problem

The presence of surfaces of vanishing influence in the space of cognitive architectures seems to be a generic feature across many model parameters. For simplicity, let us assume that the computation of influence depends solely on Δt and a critical timescale $\tau_{*}$. If the agent samples on a timescale shorter than $\tau_{*}$, they compute the influence to be $T_{12}^{<}$; otherwise, they compute $T_{12}^{>}$. This can be modeled by a Heaviside step function, Θ, so that $T_{12}^{\Delta t}=\left(T_{12}^{>}-T_{12}^{<}\right)\Theta \left(\Delta t-\tau_{*}\right)+T_{12}^{<}$. Then, the belief distribution over values of influence is bimodal:(9)$$\begin{array}{cc} \rho (T_{12}) & =\frac{1}{2}\left(1-f\right)\delta \left(T_{12}-T_{12}^{<}\right)+\frac{1}{2}\left(1+f\right)\delta \left(T_{12}-T_{12}^{>}\right)\\ \; & \mathrm{where}\;f=\mathrm{erf}\left(\sqrt{\ln\frac{\tau }{\langle \Delta t\rangle }}\cosh\sqrt{\ln{\left(\frac{\tau }{\langle \Delta t\rangle }\right)}^{1/\theta }}\right) \end{array}$$
with erf(x) the Gaussian error function. Both outcomes in the polity belief distribution are weighed by a prefactor that is determined by what fraction of the polity distribution function lies on either side of the critical sampling time.

A similar result can be found if we allow for uncertainty in the computation of influence, and polity censuring. Consider the case where an agent that has a memory less than some critical memory size $N_{1}$ computes an influence with mean $T_{12}^{1}$ and variance $\sigma_{1}^{2}$, while agents above $N_{1}$ but below $N_{2}$ compute an influence with mean $T_{12}^{2}$ and variance $\sigma_{2}^{2}$. Agents with memory above $N_{2}$ are censured so that their beliefs do not contribute to the polity. We also assume the errors that are small enough that the influence distributions are Gaussian. The belief distribution in the polity is
(10)$$\begin{array}{cc} \rho (T_{12}) & =\frac{1}{\sqrt{2\pi }}\left(\frac{1-x^{N_{1}}}{1-x^{N_{2}}}\sigma_{1}^{-1}e^{-\frac{{\left(T_{12}-T_{12}^{1}\right)}^{2}}{2\sigma_{1}^{2}}}+\frac{x^{N_{1}}-x^{N_{2}}}{1-x^{N_{2}}}\sigma_{2}^{-1}e^{-\frac{{\left(T_{12}-T_{12}^{2}\right)}^{2}}{2\sigma_{2}^{2}}}\right) \end{array}$$
(11)$$\begin{array}{cc} \; & \mathrm{where}\;x=1-\frac{1}{\langle N\rangle } \end{array}$$
For more details on either calculation, see Appendix B. Again, we find a bimodal distribution with each mode weighed by the structural and statistical properties of the polity. This is quite general, and creating distributions with more modes becomes a simple generalization.

A multi-modal polity belief distribution implies that there are *camps* within the polity with similar beliefs, though not necessarily with similar cognitive architectures. This is not a problem for consensus if the modes all have the same sign for influence—different camps still agree on the direction of influence. However, a problem occurs if the modes have different signs i.e., the distribution cuts through a line of vanishing influence. In this case, the camps have opposing beliefs, and the polity will not have a consensus belief in the direction of influence. We call this impasse *the Consensus Problem*, and it should be clear now how the heterogeneity of cognitive architectures within a polity contributes to its emergence. In the next section, we show this emergence in real world data.

## 4. An Argument over Climate Data

As an illustration of our framework applicable to empirical data, we use climate data gathered at Mauna Loa Observatory, CO${}_{2}$ content and local temperatures [43,44,45]. For clarity, we are not attempting to examine whether carbon dioxide content is driving temperature, or vice versa, but to show that the consensus problem can be identified in data coming from a dynamical system whose dynamics need not be known. These data consist of monthly measurements from 1958–present, with accuracy at the $10^{-2}$ p.p.m. and 0.1 °C levels. We leverage the uncertainty at the data level of accuracy to bootstrap a polity of heterogeneous agents with different ages. This is done by taking substreams drawn from the full data, starting dates uniformly drawn from 1958–2010, and introducing Gaussian noise with standard deviation equal to the significant digit in the original data. We experimented with many homogeneous polities of equal age agents, each instance using substreams of the same length but not necessarily starting at the present, and found the results robust for lengths up to ~20–30 months, beyond which data volume became an issue. Our bootstrapped results for a polity consisting of many ages are shown in Figure 4.

The analysis was done on the raw data, as well as detrended data, representing polities aware of long-term trends. Removing the linear/exponential trends increased the transfer entropies by nearly an order of magnitude, and tightened the error bars. Because influence is insensitive to scale transformations, the former effect did not change the overall shape of the influence curve significantly. Further detrending by the removal of the highest power harmonics had almost no effect on both transfer entropies and influence. For a more detailed description of the procedure, see Appendix C.

Figure 4B shows $T_{{T,\mathrm{CO}}_{2}}$ does, indeed, depend on the memory usage/look-back window. For NΔt<6 months, a period associated with changing weather, an agent as modeled above would infer temperature influences CO${}_{2}$ content. For NΔt>6 months, a period associated with seasonal changes, an agent would infer the opposite influence. The two data streams yield contradictory conclusions about which process influences which dependent on the architecture of the agent.

Moving up to polities, consensus on the influence direction for a random sample of agents is difficult unless the sample is drawn so that all the agents have similar memory usage given this monthly sampling strategy. The shape of the curve is similar to the second example discussed in the previous section, so we expect the polity belief distribution to be bimodal, and this is supported by Figure 5. There we see several polity belief distributions for three different values of the average polity memory size. Around 〈N〉≈6, the median value of the influence becomes 0, implying that the population is equally divided over their opinions. For populations with shorter memories, the belief that temperature is influencing CO${}_{2}$ is dominant. Populations with longer memories tend to believe the opposite. Once again, we intend this as a demonstration of how the consensus problem might emerge in polities, and understand that the complexities of weather and climate change cannot be boiled down to two simple data streams.

## 5. Discussion and Conclusions

Under the assumption that information is physical; that any realistic epistemic agent will have bounds on its ability to acquire, store, and process information from the environment stemming from its finite cognitive architecture [26,32,46,47,48,49,50,51,52,53], this paper has given a powerful reinterpretation of known problems with transfer entropy estimation as a source for disagreement within populations of such agents that span a large enough volume of possible cognitive architectures. The consensus problem does not stem from any differences in the sensory data different agents are exposed to; exposing agents to identical data streams does not ensure that all will reach the same conclusion. This result is qualitatively obvious to anyone familiar with large groups of humans and has implications for recent studies on opinion formation and polarization [54,55,56]. The results also have implications for presumably model-agnostic machine learning. ML architectures utilize information theoretic measures to *learn* from data. However, such learning demands efficient storage of belief distributions in lieu of enormous data sets and is therefore subject to historical sampling. Algorithms with memory usage optimized to specific hardware architectures will likely encounter the consensus problem described here when compared with identical algorithms running on different architectures.

Transfer entropy has gained in popularity recently in the analysis of group dynamics [57,58,59]. We hope to investigate whether our results hold for more than two data streams, and what happens as we attempt to move up in scale from small groups of agents to flocks and coarse grained polity? Extending this work to larger scales will help clarify the dynamics of group formation and models of interaction in social organisms [60,61]. Beyond that, how does the consensus problem manifest due to cognitive architectural choices in the inference of other information theoretic measures? Going back to the notion of agency, epistemic agents have a repertoire of actions available to them which they must choose from based on the conclusions they make about their changing environment: In what way does this repertoire affect the availability of sensory data to an agent introducing new ways for the consensus problem to manifest. These points open up the need to clarify what is meant by an epistemic agent, as well as a further development of the notion of a polity as a group of agents. 

## Figures and Tables

**Figure 1 entropy-24-01378-f001:**
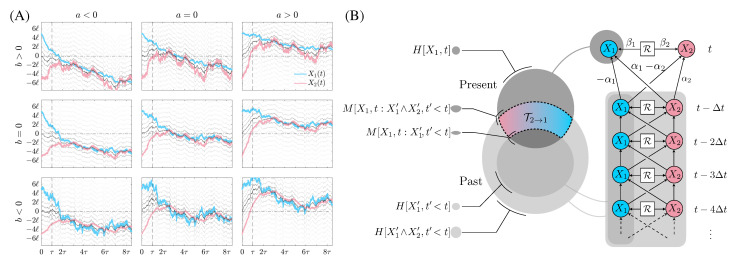
(**A**) Instances of paths generated by equations of motion, Equations (Equation 4) and (Equation 5), with the same seed for each pseudorandom generator used to simulate the heat bath. Axes have been scaled to the natural length and time scales, *ℓ* and τ, respectively. Paths are initialized 10*ℓ* units from each other. Note the existence of transient behavior decaying with timescale τ followed by steady state behavior dominated by stochasticity. The black line represents the mean of the two processes while the grey lines of decreasing opacity are an integer number of *ℓ*s away from the mean; (**B**) an information diagram of transfer entropy in our toy model. To the right, we have our coupled stochastic processes $X_{1}$ and $X_{2}$, as well as the heat reservoir, R, through which noise is introduced to both processes. Coupling constants are labelled $\alpha_{i}$ and $\beta_{i}$. The *present* is at time *t*, and the temporal discretization scale is Δt. To the left, we have an information diagram where each circle represents the entropy of that quantity. The present and past entropies are shown as bubbles, and the mutual information is the intersection of these bubbles. The transfer entropy is labeled in relation to this mutual information. Note that this diagram has a mirror image on the other side of the processes (not shown) that represents the reverse $T_{1\to 2}$ calculation.

**Figure 2 entropy-24-01378-f002:**
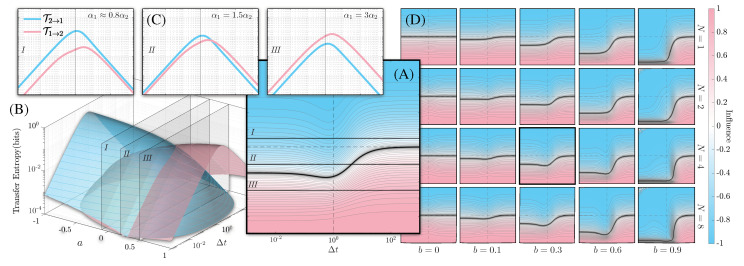
(**A**) The influence contours in our toy model across a∈[−1,1] and $\Delta t\in [10^{-3},10^{3}]$ for b=0.03 and N=4. The three slices, I,II, and III, are taken at a=−0.1,0.2,0.5, respectively. (**B**) The transfer entropy surfaces from which (**A**) was constructed, with slices shown. (**C**) The three slices are displayed. Note the crossover for a=0.2. (**D**) $T_{12}$ is plotted for other values of *b* and *N*, with a vanishing influence represented by the black line. Each contour plot has the same axes as (**A**). For negative values of *b*, the diagrams would have a flipped color scheme. Note how for constant *a* values the sign flip in the asymmetry of transfer entropy is a generic feature across a large portion of the parameter space, typically occurring within an order of magnitude of the natural timescale.

**Figure 3 entropy-24-01378-f003:**
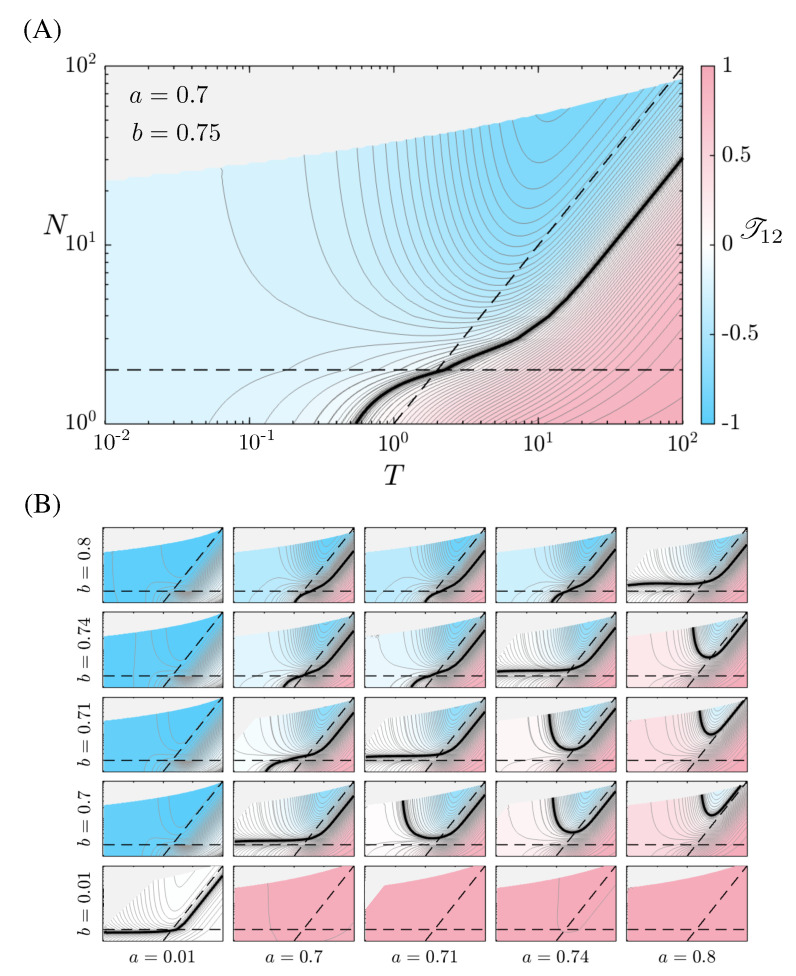
(**A**) Influence between processes with a=0.7 and b=0.75. Due to numerical accuracy, the grey regions have been removed from the plot. The horizontal line represents N=2, while the diagonal line represents look-back windows that have been cut into intervals of duration equal to the natural timescale of the system, Δt=1 (in dimensionless units). The black curve cutting through the plot is null information flow. For constant look-back window size, *T*, the direction of information flow depends on the discretization of the window; (**B**) a plot array for several values of the deterministic and stochastic parameters. Each subplot has the same description as the top panel. Note that the sign cross-over is a generic feature in much of the parameter space; furthermore, it is typically found within an order of magnitude of the natural timescale.

**Figure 4 entropy-24-01378-f004:**
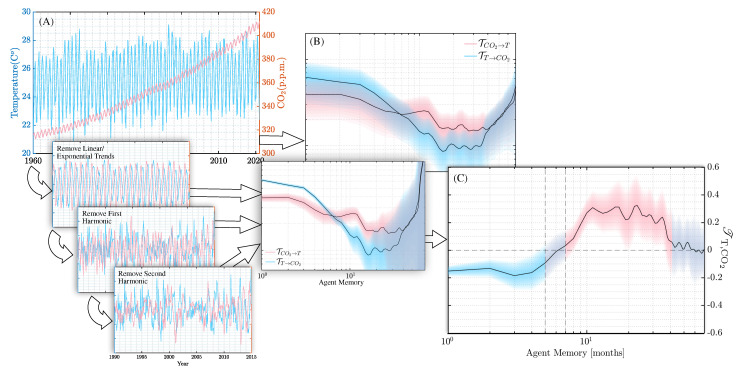
(**A**) Raw data from temperature and CO${}_{2}$ content sensors taken at Mauna Loa from 1958–2022. Smaller inset plots show detrending procedures applied to the data: First, the exponential and linear trends are removed using a best fit. This is followed by the progressive removal of the harmonics containing the most signal power; (**B**) the transfer entropies computed from the pairs of data streams as a function of agent memory size. The scale on the vertical axis is irrelevant, as any scaling is removed in the next step. The progressively detrended data results in qualitatively similar transfer entropies compared to the raw data, with a noticable decrease in variance and a scaling by an order of magnitude after the initial detrending step; (**C**) the influence between the data streams—since the influence is insensitive to scale changes in the transfer entropy, both raw and detrended data result in nearly identical curves.

**Figure 5 entropy-24-01378-f005:**
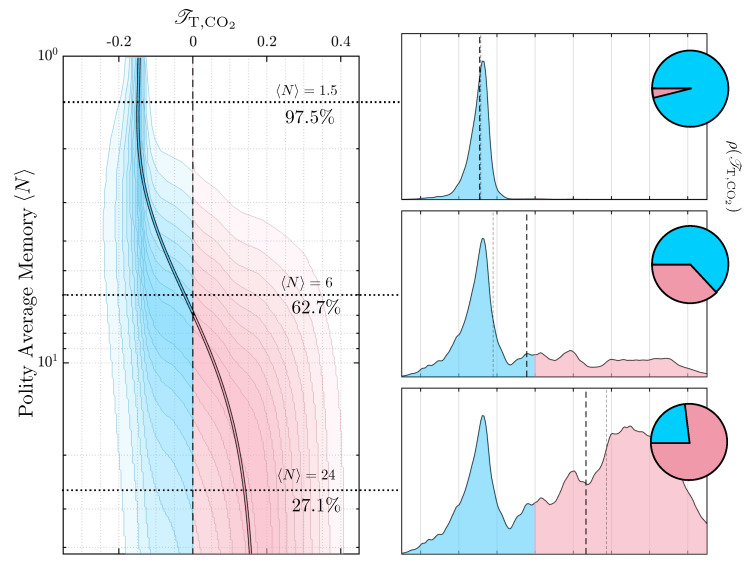
The left panel shows all the polity distributions for ensembles of agents with average memory 〈N〉. The central curve is the average value of the influence, and the margins are taken at 10% intervals centered on the median. The right panels are the belief distributions for, starting at the top, 〈N〉=1.5, 6 and 24. The thick dashed line is the mean, and the thin dashed line is the median. The inset pie charts are the fraction of the population that believe one way or the other.

## Data Availability

The data for atmospheric CO${}_{2}$ content and temperature for the Mauna Loa observation site are available at the National Oceanic and and Atmospheric Administration website [43] and the World Meteorological Organization website [44]. The code used to generate the data on influence can be shared by the corresponding author under reasonable request.

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
