# Peer review of "The Consensus Problem in Polities of Agents with Dissimilar Cognitive Architectures"

_entropy, 2022, doi:10.3390/e24101378_

Round 1

Reviewer 1 Report

The manuscript tries to make an argument about transfer entropy, i.e., that TE (or Granger causality for the case of Gaussian noise) is not a measure of causality, and it depends on the history one takes into account. The authors hide this statement under a poetic title of agents with dissimilar cognitive architectures. There are the following issues with the manuscript:

1) the title and motivation are misleading. The authors' motivation is to show that TE or GC is not causality measures. But this is not the mainstream interpretation of TE. Even Granger himself criticized the terminology of 'causality'; for TE, this approach was recently criticized, e.g., in  PRL 116,238701. This has also been discussed in terms of networks of information flows (https://doi.org/10.1142/S0219525908001465). Going one step further and considering TE as a way how an agent processes information is even more misleading. TE never had the ambition to be such a measure, its main purpose is to be a model-free measure of lagged correlation (in a general sense, not necessarily Pearson's correlation)

2) Using the net information flow (Eq. (3)) is problematic for several reasons. a) The measure does not reveal whether there is a strong bidirectional relations ships between the variables or whether they are statistically independent. b) In the case of weak correlations, it is notoriously hard to estimate the quantity since we are dividing very small quantities, and in the case of strict independence, the quantity is not even properly defined. c) As a consequence, the authors do not present any reasonable estimators of the quantity. Note that an estimator that is good for TE does not have to be a good estimator of the net normalized flow. This is the main problem, especially for the analysis of the time series in the second part of the paper.

3) Calling a set of two coupled linear differential equations with noise a model for qualia is a bit too ambitious. Calling it an agent-based model is also an overstatement. Nevertheless, the authors calculate the TE approximate value of TE for this model and conclude that TE is not a measure of causality. Again, this interpretation is not mainstream, and I dare to say that most scientists are aware of this fact.  Its main advantage is the fact that it is model-free and directly applicable to a set of time series without consideration of a particular model, as in the case of parametric measures.

4) the last point focuses on the application of TE to climate data. First, it is clear that besides temperature and CO2, there are many other factors influencing the climate, and therefore it does not make any sense to consider TE as a causal measure. Second, considering the dataset's size and the alphabet (binning) of the data, the estimates of TE for long memory time N cannot be reasonable. There are simply not enough data points to get reasonable estimates because the number of possible states of the joint distribution of the source and target variables grows as |aplhabet|^N. As a side point, the authors did not show the stationarity of both series which is a necessary property for using TE (or otherwise, detrending methods should be used).

All in all, while I can see some potential in the exploration of the parameter space of the toy model and its relation to TE, the aforementioned points are so serious that I cannot recommend the paper for publication.

Author Response

We thank the reviewer for taking the time to help improve our manuscript. There appears to be a misunderstanding of what the manuscript intends to demonstrate. We were remiss in not making the point clearer and have modified the text to ensure that readers are not mislead into thinking that the relationship between causality and transfer entropy is at the heart of the work.  We want to show how the imperfect estimates of information measures made by physical agents has consequences for groups of heterogeneous agents. The breakdown of consensus in such groups can be traced back to the heterogeneity of cognitive resources available to each agent. We have added a subsection on polities in order to quantify our assertions about the consensus problem, working through 2 examples that transition between the toy model and the results coming from the simple real world data. We have added a fifth figure to illustrate our intent better.

-Our intent is to show that agents infer, in contradiction to one another, an information measure based on their physical limitations, and what this means for ensembles of agents. 
-Estimating the net information flow with fidelity would be interesting, but not in a paper illustrating how the consensus problem emerges due to the heterogeneity of the imperfect agents attempting to infer an information measure that make up a polity.
-To a sensor measuring temperature and pressure, the world IS those two measurements. A more realistic agent would have many data streams synthesized into qualia and this is an interesting idea to pursue, but not here. We have added several sentences to introduce our model better, and have changed the language to de-emphasize the TE-causality argument.   -The authors firmly believe in climate change, and are well aware that it is far more complex than two streams of measurements, as are nations and the humans making them up. We will explore more complex models of agents, more complex models of polities, and more complex datasets, but these are beyond the scope of this manuscript.    -The stationarity problem has been fixed. We added details to our analysis concerning the detrending of the data, and incorporated them in figure 4, finding that the general shape of the normalized TE curve is insensitive to multiple levels of detrending.

We hope the changes address the referee’s concerns and thank them for their critique, the addressal of which has improved the clarity of the manuscript. 

Reviewer 2 Report

Dear Authors,

the idea of investigating the notion of causal influence of TE very interesting and the paper is, globally, well written.

In my opinion the major issue is a lack in explanation when you switch from the toy model to the real data. Probably a figure when you show the real data should be informative.

Then the properties that you investigated in your toy model should be discussed for the real case, too.

MINOR ISSUES

- Can you clarify the role of R in figure 1?

- line 238: wioth -> with

Author Response

Thank you for taking the time to review our manuscript.

In response to your comments: R represents a generic heat reservoir - the noise introduced by a complex environment on the dynamics of the two processes. We have added a sentence to clarify this. Furthermore we have added several sections on polities - moving up from a single agent to an ensemble of agents. Some examples are worked through, and only then is the analysis on real data done.

To clarify, we are not attempting to show any sort of causal relationship between CO2 content and temperature; we use the data to showcase how finite agents can reach different conclusions which leads to the consensus problem in heterogeneous polities.  

We feel that the new sections do this better, and make the manuscript more cohesive. We hope that the reviewer feels the same. 

Reviewer 3 Report

The authors discuss the problem of entropy transfer between dynamical systems. Generally, the paper is well written and the subject adequately presented. The only doubts steam from the relationship between the toy model and the real-time series analysed. I think the authors could also give an analysis of the example with the dynamics similar to the theoretically analysed problem.

Author Response

We thank the reviewer for taking the time to help us improve our manuscript.

To make the transition from toy model to real data more palpable, we have expanded the space between them by adding a quantitative subsection on polities. We take the results from the toy model and use them to model two examples which predict the emergence of bimodality in the polity belief distribution i.e. the emergence of the consensus problem. We have added a more complete presentation of our data analysis of the CO2/Temperature data to show how robust the resulting influence curve is. The examples discussed prior to this have similar shapes to the real data, facilitating the transition from the toy model to real data. Finally we find the polity belief distribution for a wide range of heterogeneous polities and show when said heterogeneity leads to the consensus problem. 

We believe that the manuscript feels significantly more complete with these additions, and hope the reviewer does as well. Once again, many thanks. 

Round 2

Reviewer 1 Report

I am still not sure about the novelty of the manuscript; however, the authors corrected the main points of my criticism, and the current version of the manuscript is improved enough that the manuscript can be accepted for publication in Entropy.